# Pilot Study Exploring the Perspectives of Canadian Clients Who Received Digitally Delivered Psychotherapies Utilized for Trauma-Affected Populations

**DOI:** 10.3390/ijerph22020220

**Published:** 2025-02-04

**Authors:** Sidney Yap, Rashell R. Allen, Katherine S. Bright, Matthew R. G. Brown, Lisa Burback, Jake Hayward, Olga Winkler, Kristopher Wells, Chelsea Jones, Phillip R. Sevigny, Megan McElheran, Keith Zukiwski, Andrew J. Greenshaw, Suzette Brémault-Phillips

**Affiliations:** 1Department of Psychiatry, Faculty of Medicine and Dentistry, University of Alberta, Edmonton, AB T6G 2H5, Canada; syap@ualberta.ca (S.Y.); burback@ualberta.ca (L.B.); ow@ualberta.ca (O.W.); agreensh@ualberta.ca (A.J.G.); 2Heroes in Mind, Advocacy, and Research Consortium, Faculty of Rehabilitation Medicine, University of Alberta, Edmonton, AB T6G 2G4, Canada; kbright@ualberta.ca (K.S.B.); mbrown2@ualberta.ca (M.R.G.B.); cweiman@ualberta.ca (C.J.); psevigny@ualberta.ca (P.R.S.); 3School of Clinical Child Psychology, Faculty of Education, University of Alberta, Edmonton, AB T6G 2G5, Canada; wozniak@ualberta.ca; 4School of Nursing and Midwifery, Mount Royal University, Calgary, AB T3E 6K6, Canada; 5Department of Occupational Therapy, Faculty of Rehabilitation Medicine, University of Alberta, Edmonton, AB T6G 2G4, Canada; 6Department of Computing Science, Faculty of Science, University of Alberta, Edmonton, AB T6G 2E8, Canada; 7Department of Emergency Medicine, Faculty of Medicine and Dentistry, University of Alberta, Edmonton, AB T6G 2T4, Canada; jhayward@ualberta.ca; 8Department of Child and Youth Care, Faculty of Health and Community Studies, MacEwan University, Edmonton, AB T5J 4S2, Canada; kristopher.wells@macewan.ca; 9Faculty of Education, University of Alberta, Edmonton, AB T6G 2G5, Canada; 10Wayfound Mental Health Group, Calgary, AB T2R 1J5, Canada; meganm@wayfound.ca; 11Haikei Health, Edmonton, AB T5W 3H1, Canada; keith@drzukiwski.com

**Keywords:** digital mental health interventions, mixed-methods study, trauma-focused psychotherapy, military members and veterans, public safety personnel

## Abstract

The digital delivery of mental health services became increasingly common following the onset of the COVID-19 pandemic. There is still much to learn regarding tailoring interventions for trauma-affected populations (military members, Veterans, public safety personnel). Through the current pilot study, we explored the perceptions of digitally delivered psychotherapies utilized for trauma-affected populations, as reported by Canadian military members, Veterans, and public safety personnel who completed such interventions. Quantitative data were collected from 11 Canadian clients (military members, Veterans, and public safety personnel with posttraumatic stress injury). Survey questions were based on the Alberta Quality Matrix of Health and the Unified Theory of Acceptance and Use of Technology model. As a follow-up, clients were invited to partake in a semi-structured interview to further explore their perspectives on digitally delivered trauma-focused and adjunct therapies. Four clients participated in an interview. The client participants reported that digitally delivered trauma and adjunct therapies offered similar treatment effectiveness to in-person delivery while also improving treatment access. The participants indicated several unique advantages of digital delivery, including the increased accessibility of treatment, cost-effectiveness, and more efficient use of resources, although the small sample size limits the generalizability of our findings. Further research with a larger, more diverse population is required to corroborate our results and identify other avenues in which psychotherapies utilized for trauma-affected populations can be engaged with and improved upon.

## 1. Introduction

The COVID-19 pandemic substantially impacted approaches to service delivery and mental health. In Canada, more than 40% of polled individuals reported experiencing mental distress early in 2021 when COVID-19-related precautions were commonplace [1]. The pandemic further stressed groups at high risk for potentially psychologically traumatic event exposures (PPTEs, e.g., exposure to actual or threatened death, serious injury, or sexual violence) [2,3]. Such groups include public safety personnel (PSP) and frontline healthcare workers, as well as currently serving military members and Veterans [3,4]. Potentially psychologically traumatic event exposures can have physical impacts (e.g., digestive and cardiovascular diseases), result in behavioral challenges (e.g., maladaptive responses like burnout, alcoholism, and suicidality), and induce posttraumatic stress injuries (PTSIs, e.g., posttraumatic stress disorder [PTSD] and major depressive disorder) [2].

A pre-COVID-19 study found that 44.5% of Canadian PSP self-reported a mental disorder diagnosis [4]. Further, 23.2% of this Canadian PSP population yielded a positive screen for PTSD. Public safety personnel also exhibit elevated psychopathology despite prior treatment, indicating clinical complexity and the potential need for transdiagnostic, specialized, or ongoing mental health support [5]. Canadian Armed Forces (CAF) Regular Force Veterans exhibited a similar prevalence of PTSD (16%) to PSP [6]. Additionally, this population reported a mood disorder prevalence of 21% and an anxiety disorder prevalence of 15% [6].

The unique mental health challenges experienced by PSP, military members, and Veterans may have been compounded by the COVID-19 pandemic [7]. According to the Public Health Agency of Canada, those who met the criteria for PTSD during the pandemic were more likely to report being impacted by the pandemic (and mandatory public health restrictions and protocols) in terms of physical health problems, difficulties in meeting financial obligations/essential needs, and challenges faced in personal relationships with household members. Those individuals were also more likely to report symptoms of anxiety and depression within the past two weeks, greater lifetime suicidal thoughts, and increased use of alcohol and cannabis [8].

Transformation in service delivery was prompted by the escalation of mental health concerns due to COVID-19 and the need to adhere to physical distancing mandates and government restrictions. The pandemic required an unprecedented rapid transition from in-person to digital methods (e.g., teletherapy, telemedicine, eHealth, and mobile health) [9]. Considering the mental health consequences of the COVID-19 pandemic [1,10], digital mental health interventions (DMHIs) offered a cost-effective alternative to in-person mental health treatment while complying with public health requirements for physical distancing during the pandemic [11]. Such interventions allowed for access to timely and secure trauma therapies that are critical to supporting the transdiagnostic needs of PSP, military members, and Veterans [12]. Accordingly, the COVID-19 pandemic greatly accelerated the uptake of DMHIs [13].

Research indicates that digitally delivered Prolonged Exposure (PE), Eye Movement Desensitization and Reprocessing (EMDR), and Trauma-Focused Cognitive Behavioral Therapy (TF-CBT) may be effective at reducing PTSD symptomatology [14,15]. DMHIs generally provide clients with increased convenience, comfort, and access to treatment [16,17]. The increased access to care can lead to substantial cost and time savings for clients [11]. DMHIs may also aid in decreasing the stigma related to accessing mental healthcare [18].

Evidence supporting the wide-scale use of DMHIs for clients experiencing PTSIs, despite promising findings supporting the use of DMHIs for general mental healthcare, is comparatively scarce. Few randomized controlled trials (RCTs) have directly compared the effectiveness of digitally delivered and in-person trauma interventions [19]. Some Veteran clients have reported that DMHIs were negatively impacted due to Internet connection issues and that therapy delivery felt distant and impersonal in the digital environment [19]. Those with greater symptom severity may also be less likely to complete treatment programs delivered digitally, as evidenced in a recent study evaluating the use of Internet-Delivered Cognitive Behavioral Therapy (ICBT) in PSP [20]. Such studies have been comparatively rare, however, with much of the extant DMHI literature primarily focusing on the use of DMHIs in civilian populations, such as the use of ICBT in civilian populations [21,22]. This has limited our ability to understand how the shift to digital delivery has affected PPTE-focused treatments specifically. There is clearly a need to explore the mitigation of the potential risks and disadvantages of using digital trauma interventions for those experiencing PTSIs, including appropriately assessing client compatibility, obtaining consent in a secure manner, ensuring safety in the context of suicidal ideation or behavior, adapting therapy protocols to the digital environment, and ensuring ongoing client privacy [14].

Most current research into the use of DMHIs for the purpose of trauma-focused treatment centers almost exclusively on military and Veteran populations in the United States. This geographic focus highlights a critical lack of research specifically examining the experiences of Canadian PSP, military members, and Veterans who have received digitally delivered psychotherapies utilized for trauma-affected populations. Multiple differences between Canadian and American client populations, including differing social characteristics (e.g., different cultural backgrounds), healthcare needs, and geographical challenges, may substantially impact how these populations access and utilize DMHIs.

The use of digitally delivered therapies is becoming increasingly common and rapidly evolving to include virtual reality-based interventions [23,24]. Against this backdrop, we conducted this pilot study to investigate the perceptions of synchronous therapist-delivered digitally delivered psychotherapies utilized for trauma-affected populations, as reported by Canadian military members, Veterans, and PSP who have undergone these interventions. To understand the strengths and weaknesses of these interventions, surveys adapted from the Alberta Quality Matrix for Health (AQMH) [25] and the Unified Theory of Acceptance and Use of Technology (UTAUT) [26] and semi-structured interviews were utilized. The state of implementation of DMHIs from the perspectives of clients, clinicians, and community partners was reported on previously [27].

## 2. Methods

The current study used a mixed-methods design [28] in a community-engaged research setting [29]. Client participants completed a battery of surveys administered using Research Electronic Data Capture (REDCap 14.5.2), a secure web application used for building, managing, and completing online surveys and databases [30,31,32]. This battery included a demographic survey and surveys based on the AQMH and UTAUT. Participants were invited to participate in a 30–60 min semi-structured interview, conducted over Zoom (version 5.15), to further explore their perspectives on psychotherapies utilized for trauma-affected populations.

Study data were stored on the REDCap server and a dedicated, encrypted, and password-protected research drive hosted by the Faculty of Rehabilitation Medicine at the University of Alberta only accessible by research team members. Ethics approval was obtained from the University of Alberta’s Health Research Ethics Board (Pro00109065) prior to the commencement of research activities. Written informed consent was obtained from all study participants prior to engaging in research activities.

### 2.1. Participant Inclusion and Exclusion Criteria

Currently serving military members, Veterans, and PSP who were receiving or who had received synchronous therapist-provided digitally delivered trauma therapy from a mental health clinician in Canada, either through an Operational Stress Injury Clinic (military member and Veteran clients only) or a private provider, were recruited. All client participants had a current or prior diagnosis of a PTSI (e.g., PTSD), which may have stemmed from operational injuries or past adverse experiences (e.g., adverse childhood events). Individuals were excluded from the study if they were not a current or former military member, Veteran, or PSP; if they had not received digitally delivered trauma therapy from a mental health clinician in Canada; and/or if they did not have a current or prior PTSI diagnosis. Individuals who were under 18 years of age, and/or unable to provide informed written consent, and/or not fluent in English were excluded from the study. Non-English-speaking individuals were excluded due to the research team’s limited capacity to conduct interviews and analyze data in multiple languages.

The same inclusion and exclusion criteria were used in a previous publication [27].

### 2.2. Recruitment and Data Collection

Snowball and purposeful sampling strategies were used to recruit participants from partner and non-partner mental health clinics. These clinics were provided with study recruitment materials and information, which were passed on to potential client participants. Interested participants who completed a consent-to-contact form over REDCap were contacted by a member of the research team by telephone or email to discuss the study, determine eligibility, and assess their willingness to voluntarily participate. Recruitment took place between January 2022 and March 2023.

A link to a REDCap webpage was shared with eligible and interested individuals to access and complete the informed consent form. Participants who provided informed consent completed survey measures and/or a semi-structured interview. Surveys were iteratively developed based on the AQMH [25] and UTAUT [26] by the research team. These measures were crafted to maximally align the survey language with each AQMH and UTAUT dimension, integrating the principles of equity, diversity, and inclusion; minimizing survey burden; and maximizing data quality, collection, and analyses. Data collection took place from February 2022 to May 2023.

### 2.3. Tools and Measures

#### 2.3.1. Survey Based on the Alberta Quality Matrix for Health Survey

The AQMH [25] was created in 2005 based on the work of Crossing the Quality Chasm: A New Health System for the 21st Century [33]. The AQMH can be used to organize information about complex health systems for analyses and create awareness of quality in service delivery. The AQMH has the following two components: (1) dimensions of quality, which focuses on aspects of the patient and client experience, and (2) areas of need, which divides services provided by the health system into four distinct but related categories (being healthy, getting better, living with illness or disability, and end of life). The components are each considered across the following six dimensions: acceptability, accessibility, appropriateness, effectiveness, efficiency, and safety [25].

The research team iteratively developed a survey based on the AQMH and the previous literature (see Appendix A for a copy of the AQMH and the survey adapted from the AQMH). The survey consisted of 10 items, scored on a 7-point Likert-type scale from 1 (strongly disagree) to 7 (strongly agree), which explored the ease of use, convenience, acceptability, practicality, accessibility, appropriateness, effectiveness, efficiency, safety, and fit of in-person and digital delivery. The conversion allowed participants to rate the quality of service of in-person and digital delivery of trauma therapies along the AQMH dimensions.

#### 2.3.2. Unified Theory of Acceptance and Use of Technology Survey

The UTAUT model was developed by Venkatesh et al. as a synthesis of eight technology acceptance models. The UTAUT was designed to assess the acceptance of new technology and may explain up to 70% of the variance in intention to use technologies [26]. The UTAUT has well-established construct and content validity. The six factors influencing technology use as measured by the UTAUT include the following:Effort Expectancy: the degree of ease associated with using the technology. If participants perceived psychotherapies utilized for trauma-affected populations to have low Effort Expectancy, it would be expected that they would be more likely to use it.Performance Expectancy: assesses whether the participant believed that the technology would improve the performance of the job they were trying to complete. If belief in psychotherapies utilized for trauma-affected populations was high, participants would be more likely to use the technology.Behavioral Intention: the degree to which participants had a conscious plan to utilize technology. This construct, in turn, predicts Use Behavior and technology acceptance.Social Influence: the extent to which individuals surrounding the participant perceived the usefulness of psychotherapies utilized for trauma-affected populations and how much these important others influenced the participant’s use of psychotherapies utilized for trauma-affected populations.Facilitating Conditions: the extent to which conditions, such as organizational and technical infrastructure, surrounding the participant support the use of psychotherapies utilized for trauma-affected populations.Use Behavior: the extent to which participants used psychotherapies utilized for trauma-affected populations.

The research team iteratively developed a survey based on the UTAUT content and components, as well as the previous literature (see Appendix A for a copy of the UTAUT and the surveys adapted from the UTAUT). This survey consisted of 18 questions, scored on a 7-point Likert-type scale from 1 (strongly disagree) to 7 (strongly agree). Each of the 6 UTAUT constructs was measured individually, with 3 questions asked per construct. All construct scores were then combined to assess the overall usability of the technology used for digitally delivered trauma therapy [26].

#### 2.3.3. Semi-Structured Interviews

Qualitative data were collected via 30–60 min semi-structured solo interviews (*n* = 4), conducted and recorded over Zoom [34]. All Zoom interviews were conducted using Zoom Business set up with end-to-end encryption and geolocation limited to servers within Canada. The aim of these interviews was to further explore client perspectives on digitally delivered psychotherapies utilized for trauma-affected populations. Interviews were facilitated by members of the study team using a semi-structured interview guide. Key topics of discussion included the previous and current state of using digitally delivered psychotherapies utilized for trauma-affected populations in the midst of the COVID-19 pandemic; barriers to, facilitators of, and recommendations for the use of psychotherapies utilized for trauma-affected populations to deliver mental health services to military members, Veterans, and PSP; issues, acceptance, and methods of delivery for psychotherapies utilized for trauma-affected populations; clinical effectiveness; and needs, including infrastructure and implementation. Participants were given the option to turn their cameras off and change their Zoom display name during the interview to maintain their anonymity.

See Appendix A for copies of interview scripts used for client interviews.

### 2.4. Data Analysis

All data were de-identified prior to data analysis. This involved removing names and contact information. De-identified survey data were analyzed using IBM SPSS Statistics software (Version 28.0) [35], while NVIVO 13 (2021, R1) was used to facilitate the analysis of de-identified interview data.

Descriptive statistics were calculated for each survey variable for client participants. Non-parametric analyses were conducted due to the relatively small sample size. Paired-sample Wilcoxon signed-rank tests were used to assess within-subject differences (*p* ≤ 0.05) in median AQMH survey dimension scores between digitally delivered and in-person psychotherapies. One-sample Wilcoxon signed-rank tests were also used to assess statistically significant within-subject differences (*p* ≤ 0.05) between observed and reference median UTAUT survey dimension scores, where the reference score was 12 (i.e., neutral score, based on the sum of 3 total questions per UTAUT dimension asked on a Likert scale from 1 to 7, where 4 is the median score for each question). In total, 10 tests were conducted for the AQMH survey results, and 6 tests were conducted for the UTAUT survey results. To address multiple comparisons across the 16 statistical tests, the Benjamini–Hochberg procedure was used to control the False Discovery Rate (FDR) [36]. A summary of statistical tests can be found in Appendix A.

Video-recorded interviews were transcribed using the automated transcription function in Adobe Premiere Pro, running locally on a secure University of Alberta computer. Transcription accuracy was checked by a research team member (SY or RW), anonymized, and thematically analyzed both deductively and inductively following an iterative process [37]. Research team members were blinded to the data during thematic analysis. Deductively, initial codes were developed based on interview topics and study objectives. Inductive coding involved identifying themes that emerged from collected data. Coding for each interview was independently conducted by two research team members (SY, RW), after which a senior researcher (SBP) reviewed and refined the codes. These were then combined and tabulated into preliminary themes. The analysis of preliminary themes by the larger research team followed, with differences being resolved through discussion. A proposed thematic theory then underwent collective analysis, where preliminary themes were modified and key quotes isolated to illustrate the selected themes. The final thematic narrative was then prepared. The Standard for Reporting Qualitative Research was used to guide the reporting process [38].

## 3. Results

### 3.1. Client Participant Demographics

Eleven Canadian military members, Veterans, and PSP completed survey measures. Client participants were an average age of 50 ± 10.5 years (range: 34 to 61 years old) and self-identified as female (*n* = 3), male (*n* = 8), women (*n* = 3), or men (*n* = 8) and as Caucasian (*n* = 9) or Aboriginal/Metis (*n* = 2). No participants identified as trans or gender-diverse. Our participants self-reported being in the CAF (*n* = 7), in the police force (*n* = 2), a paramedic (*n* = 1), or a correctional worker (*n* = 1), with an average length of service of 21 ± 9.6 years (range: 5 to 38 years). Many participants (*n* = 8) reported having received some form of psychotherapeutic treatment, including CBT (*n* = 5), CPT (*n* = 1), EMDR (*n* = 3), and PE (*n* = 2), for their trauma prior to participating in the current study. Our data, however, did not allow us to distinguish between those who had experience receiving in-person trauma therapy from those who had not received non-trauma-specific therapy previously. Four participants had received digitally delivered therapies prior to engaging in the study, while seven were receiving digitally delivered therapies for the first time during study participation. All participants had received psychotherapies utilized for trauma-affected populations via video conferencing, with some reporting having received additional services via telephone (*n* = 3).

Four client participants took part in a semi-structured interview (2 females, 2 males; 2 women, 2 men; 3 Caucasian, 1 Aboriginal/Metis). Participants self-reported being in the CAF (*n* = 2) or police force (*n* = 2) with an average length of service of 19 ± 11.9 years (range: 5 to 31 years). Two participants reported having received some form of psychotherapeutic treatment, including CBT (*n* = 2), EMDR (*n* = 2), and PE (*n* = 1), for their trauma prior to participating in the current study. One participant had received digitally delivered therapies prior to engaging in the study, while three were receiving digitally delivered therapies for the first time during study participation.

### 3.2. Survey Results

#### 3.2.1. Survey Based on Alberta Quality Matrix for Health Results

Participants rated convenience (*p* = 0.011), practicality (*p* = 0.011), accessibility (*p* = 0.012), and efficiency (*p* = 0.014) statistically significantly higher for the digital delivery of trauma therapy compared to in-person delivery. There were no statistically significant differences in client participant ratings for the ease of use, acceptability, appropriateness, effectiveness, safety, and fit dimensions between digitally delivered and in-person trauma therapies (Figure 1, Table 1).

#### 3.2.2. Survey Based on Unified Theory of Acceptance and Use of Technology Results

All client participants agreed that digitally delivered trauma therapy services were a viable alternative to in-person delivery. Client participants indicated they somewhat agreed, agreed, or strongly agreed with the Effort Expectancy (score: 16/21), Performance Expectancy (score: 17/21), Behavioral Intention (score: 18/21), and Use Behavior (score: 15/21) constructs. Multiple comparison analysis revealed that only Performance Expectancy (*p* = 0.011) scores were significantly different compared to the expected median scores. Client participants also indicated that they neither agreed nor disagreed with the Social Influence (score: 12/21) and Facilitating Conditions (score: 13/21) constructs (Figure 2, Table 2).

### 3.3. Interview Results

Thematic analysis of the interview data isolated five main themes regarding digitally delivered psychotherapies utilized for trauma-affected populations: (1) creating connection while online; (2) improved access to care; (3) differing experiences working digitally; (4) difficulties with working digitally; and (5) continuing to improve accessibility of care. See Table 3 for a summary of interview results.

#### 3.3.1. Theme 1: Creating Connection While Online

Client participants felt very comfortable receiving digitally delivered psychotherapies utilized for trauma-affected populations. This comfort reportedly stemmed from their familiarity with using digital platforms, such as Zoom, in their occupations and daily lives, which saw increased usage following the onset of the COVID-19 pandemic [39]. Being comfortable and familiar with working in digital environments made client participants feel more ready to engage with digitally delivered psychotherapies utilized for trauma-affected populations, which in turn aided in developing a strong therapeutic alliance with their clinician.


*“I didn’t because I had used it. I’ve been using it since the beginning of COVID with my work that I was doing. I felt just as close to the virtual therapist as the in-person one. I did feel a real sense of of bonding with them and feeling as if I could disclose things and talk to them[.] I didn’t find it different in that sense[.] I made a very strong connection with the therapist very quickly. She was excellent. Very experienced, and I felt very comfortable with her.”*
[Client participant 4]

#### 3.3.2. Theme 2: Improved Access to Care

Client participants agreed that the most crucial benefit of digital delivery was that it increased the accessibility of psychotherapies utilized for trauma-affected populations. Having digital therapy sessions allowed for more flexible scheduling, allowing client participants to attend sessions without sacrificing other responsibilities (e.g., not missing work or childcare responsibilities).


*“I did find it much easier to schedule, schedule around where I was, where [my therapist was]. And it was also easy for adjustments too if I needed to postpone or change or if he needed some extra time from something else. It was a lot easier to communicate [and] arrange schedules that way, which made for less [absences or appointments] missed.”*
[Client participant 1]


*“I mean, if you’re a, let’s say you’re a single mother with two kids and you know you can’t make it downtown or you can’t make it to wherever right to go and do your sessions, then you know you have to skip this one, skip that one and then eventually you’re just like, I’m not even going to bother phoning them anymore. Versus, oh, it’s 1:00 I should turn on my computer and have a quick little chat with somebody here who can help me through a lot. I think it would help a lot of people[.]”*
[Client participant 3]

#### 3.3.3. Theme 3: Differing Experiences Working Digitally

Client participants had varying experiences while receiving trauma therapy digitally. One client participant felt that they did not experience the same range of emotions when receiving digital care as they did in person. This potentially indicates that digital delivery may be blocking therapeutic content.


*“For some reason I didn’t get as emotional as [I] knew I probably would have if I’d been in person.”*
[Client participant 4]

Another client participant shared that digital delivery decreased the anxiety of receiving care, sharing that traveling to a clinic to receive in-person treatment caused them quite a bit of stress.


*“[It added] more stress [for] me going into therapy back then at the [Occupational Stress Injury] clinic, [being] over medicated [and] driving back and forth, not understanding what was going on in my head[...]”*
[Client participant 2]

#### 3.3.4. Theme 4: Difficulties with Digital Delivery

Some client participants found attending digital sessions to be resource-intensive, typically requiring a general knowledge of using digital platforms and access to a working computer, a stable high-speed internet connection, a web camera with clear picture, a quiet and private space to attend the session, and a support person nearby and available in the event of an adverse emotional response. This caused frustration among some participants.


*“He had to be able to talk to me through[,] to talk me through [going] on the Jane chat thing that we, right? He had to talk me through it over the phone, right, because I’m not[,] I’m not into all this stuff so[...] [I was frustrated because] I hate computers, data, and I was just like, “I don’t know how to operate it.””*
[Client participant 3]

Client participants also raised concerns over family members or significant others overhearing their trauma therapy sessions, raising questions regarding the security and privacy of digitally delivered services. Such challenges left some client participants feeling disconnected from their clinician and the therapeutic experience in general, leading to less effective treatment and a less intimate therapeutic relationship. For these participants, this left the impression that psychotherapies utilized for trauma-affected populations may not be an adequate replacement for in-person delivery.


*“Yes, so that was a little awkward because, you know, if somebody was in my house and I had to go to a different room, there was always the thought of “Can they hear me? Are they listening?” I mean, generally I was not in a house of somebody I couldn’t trust anyways. But, you know, those are private sessions that you want to keep to yourself.”*
[Client participant 1]


*“There [are] certain things about [my] experiences that I would prefer my family not to know about[,] that I keep to myself, right? You know, [my] five year old is sitting on the couch right now while we’re doing this. So, you know, that’s a little bit of a negative.”*
[Client participant 3]

#### 3.3.5. Theme 5: Continuing to Improve Accessibility of Care

Client participants indicated several recommendations for integrating digital delivery into psychotherapy care for trauma-affected populations, including increasing the opportunities for and the spread of hybrid care, a combination of digital and in-person therapeutic services.


*“Yeah, I think maybe something like you’re offering them both and if people are selecting virtual because they can get in quicker, letting them know that it could be transitioned [by] agreement or in particular what the therapist thinks would be best if you could transition into in-person somewhere down the road during the therapy. If that’s [what] looks [to be] needed. You wouldn’t get absolutely stuck with one if [you] chose it. Similarly, you could switch from in-person to virtual, maybe as you’re coming to the end of your therapy.”*
[Client participant 4]

Taken together, the survey and interview data indicate that clients believe that digitally delivered psychotherapies utilized for trauma-affected populations have several unique benefits, including better accessibility to treatment and greater client autonomy, all while offering similar therapeutic outcomes to in-person delivery. Many factors, including potential security and safety concerns, were concerning for study participants. Further investigation is warranted to address the concerns and the recommendations made by client participants.

## 4. Discussion

The current study provides preliminary evidence regarding the use of digitally delivered psychotherapies utilized for trauma-affected populations from the perspectives of Canadian military members, Veterans, and PSP who have experienced trauma. Client participants reported that digitally delivered psychotherapies utilized for trauma-affected populations appeared to offer similar treatment quality of care to in-person delivery while also improving treatment access.

The current study is one of a limited number of studies within an emerging field that lacks substantive research. Research specifically examining the experiences of Canadian clients who have received digitally delivered psychotherapies utilized for trauma-affected populations is relatively rare. Canadian clients face many unique challenges, including potentially having to travel long distances from remote regions to urban centers to access evidence-based specialized trauma treatment. Although the sample size for this study was smaller than expected, the insights shared by participants add valuable perspectives to the extant literature from a population that has historically been relatively under-researched. Further, this study provides an update on previous research in the field [12], enabling a better understanding of shifts in the attitudes toward and usage of psychotherapies utilized for trauma-affected populations over the COVID-19 pandemic. Our study also provides insight into the implementation of new DMHIs, especially in the Canadian context, highlighting the importance of co-designing services with the specific needs, such as treatment accessibility and appropriateness, of clients from typically under-researched populations in mind. Our research may provide valuable insights when considering the expansion of DMHI services within the general healthcare system.

Data from the survey based on the AQMH indicated that client participants found no statistically significant differences in service quality between digitally delivered and in-person trauma therapies. This is consistent with previous research suggesting that equivalent quality care is attainable via digital or in-person service delivery modalities [14,15]. Similar results were found in client interview data. In addition, client participants rated the convenience, practicality, accessibility, and efficiency of digitally delivered psychotherapies utilized for trauma-affected populations higher than in-person delivery. These results corroborate the findings of previous research in the field, which have identified cost-effectiveness, time savings, and increased access to therapy compared to in-person care as unique advantages of digital delivery [14,19]. Based on our findings, these dimensions of healthcare appear to be highly valued by clients receiving such therapies. Further research is yet needed to verify our results and better understand whether these dimensions of healthcare ought to be prioritized within healthcare programs.

Client participants’ UTAUT survey results indicated general agreement that digital delivery was highly usable based on relatively high scores on the Effort Expectancy, Performance Expectancy, Behavioral Intention, and Use Behavior constructs. The results also indicated that Social Influence and Facilitating Conditions did not appear to be as important to client participants, potentially indicating that client participants were willing to use psychotherapies utilized for trauma-affected populations despite a lack of perceived support from important others (e.g., family) or the organizations they work for (e.g., military organizations). Perhaps this points to safety and treatment needs overriding the need for social and organization support with regard to using DMHIs, including psychotherapies utilized for trauma-affected populations. Finally, although there are moderators known to influence Behavioral Intention and overall technology acceptance, due to the limited sample size, the influence of these moderators was not evaluated.

Despite the generally positive feedback provided by client participants, they shared some concerns regarding accessing the appropriate technology and possessing sufficient technological literacy needed to receive digitally delivered services. Similar findings were found in a qualitative study conducted in the United States that recommended that, among other factors, providing equitable device distribution and digital literacy training for clients would be necessary for the successful implementation of DMHIs [40]. Further research is needed to better understand whether these factors should be prioritized when implementing DMHIs, including psychotherapies utilized for trauma-affected populations.

Another consideration that should be made moving forward is the formation of the therapeutic relationship and managing client distress in the digital environment. Some client participants expressed difficulties in forming a strong therapeutic relationship and emotional connection with their clinician, in line with previous research that has indicated that individuals who have experienced interpersonal trauma may have difficulties developing a strong therapeutic alliance [41]. Future research could consider collecting longitudinal data on perceptions of the therapeutic alliance using instruments, such as the Brief Revised Working Alliance [42], to track how perceptions of the therapeutic alliance change over time.

A third concern shared by participants regarding DMHI was the potential lack of privacy and security while attending digitally delivered sessions. For example, some participants found it difficult to find an area in their home to attend psychotherapy sessions that was sufficiently removed from their families. This made it difficult for them to openly speak about their traumatic experiences, with their worries of conveying information regarding their traumas overtaking their ability to focus on their psychotherapy session. These ethical concerns substantially negatively impacted their therapeutic experience, making them feel disconnected from their session and unsure whether DMHI could provide an adequate replacement for in-person care. Further research is needed to better understand how to adequately address these ethical concerns.

Client participants supported the increased and expanded usage of hybrid services, where they could attend digital or in-person sessions depending on their needs. Hybrid care models could be used to address some of the issues surrounding DMHIs, including privacy and technology barriers, assisting clients in receiving the most optimal services for their mental healthcare needs. Practical steps, such as the expansion of clinical training programs focused on providing hybrid care and the installation of accessible and reliable infrastructure for therapists and clients, can be taken to ensure the successful implementation of hybrid care models. Moving forward, it will be imperative that mental health clinicians are prepared and able to identify which clients with what symptoms are most appropriate for digital, in-person, or hybrid services, which could aid in reaching positive treatment outcomes. Future research should therefore prioritize identifying which individuals would benefit most from such interventions.

### Limitations

There were several study limitations. First, all recruitment and surveys in the current study were in English, which limited responses from non-English speaking communities. Second, the sample size and the diversity of our study population were relatively limited, which precluded analyzing certain mediating effects. For example, although there are moderators known to influence Behavioral Intention and overall technology acceptance, due to the limited sample size, the influence of these moderators was not evaluated. Similarly, potential gender, sex, and racial differences were not explored, given the limited sample size. Such factors can impact how individuals interact with the mental healthcare system, as gender [43,44] and racial minorities [45] may be disproportionately affected by mental health challenges. Steps were taken in this mixed-methods study to mitigate the effects of the small sample size, including the use of rigorous and rich qualitative data collection to enhance the credibility and depth of findings. Purposive sampling was used to ensure diverse perspectives, while integrating qualitative and quantitative results strengthened interpretability. Comparing findings with the existing literature and providing thick descriptions improved transferability, while transparently acknowledging limitations reinforced methodological rigor. While these strategies aided in mitigating sample size constraints and enhanced the study’s contribution, further research with larger sample sizes is needed to replicate the findings reported here and to explore potential moderating factors, such as differences related to gender, sex, and sexual orientation with regard to the acceptance of digitally delivered care.

A third limitation of the current study was the potential influence of selection bias, as has been described in previous studies [46,47]. As we sought participants from across Canada, all recruitment and data collection were completed through an online platform. As research activities took place over online mediums, those without access to technology were excluded due to feasibility reasons. We acknowledge that the use of this digital platform may have increased the likelihood for clients already comfortable with using or more accepting of technology to enroll in our study. Additionally, we recruited clients who had successfully completed a course of digitally delivered trauma therapy, which may have impacted their overall agreeableness in relation to psychotherapies utilized for trauma-affected populations and DMHIs in general. Further, all participants had the resources required to participate in this study, including technology, internet access, and time. Evidence suggests that those from higher socioeconomic backgrounds may be less likely to experience health issues, including psychological distress, compared to individuals from lower socioeconomic backgrounds [48]. This limits the generalizability of our findings, as our study population was not representative of the populations living in remote regions of Canada, nor did it fully capture the diverse populations of Canada. Differences in geographic location may also influence the willingness of individuals to use digital or in-person care. For example, the variability in the availability of TFP resources between rural and urban centers may affect clients’ perceived preferences for using digitally delivered or in-person services. The provision of technology to those living in rural centers or the expansion of training programs for clients from diverse socioeconomic backgrounds may aid in mitigating some of the currently identified barriers to accessing DMHIs. Further research is needed to identify equitable and sustainable strategies for providing digitally delivered psychotherapies utilized for trauma-affected populations that acceptably meet the needs of Canada’s varied populations.

A fourth limitation of this study was that we were able to collect only limited information regarding the specific psychotherapeutic interventions that the client participants had received. Future research should prioritize collecting data such as the specific interventions received and the length of psychotherapy attendance to clarify whether or not participants were completing the therapeutic process.

## 5. Conclusions

Client participants shared many unique benefits and barriers to receiving digitally delivered psychotherapies utilized for trauma-affected populations. Given the high rate of PTSIs within trauma-affected populations, it is critical that they have access to the highest-quality interventions in a secure, cost-effective, and accessible manner. Our results suggest that digital delivery offers an accessible and practical way for Canadian military members, Veterans, and PSP to receive trauma therapy. This pilot study adds to the growing body of evidence supporting the use of DMHIs in trauma-affected populations and provides a canvas for future research to be built upon. Ultimately, the livelihoods of trauma-affected populations may be directly impacted and improved with the use of DMHIs.

## Figures and Tables

**Figure 1 ijerph-22-00220-f001:**
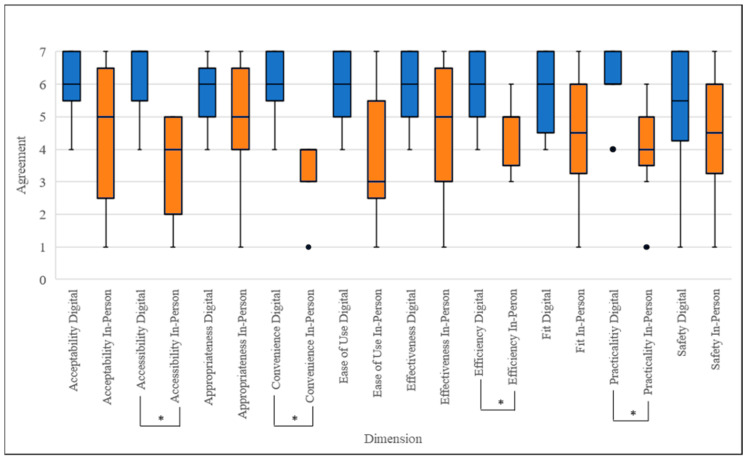
Box and whisker plots indicating client participants’ (*n* = 11) median AQMH survey scores, first and third quartiles, and minimum and maximum scores. * = Significant difference (*p* < 0.05) between median scores for digital delivery vs in-person therapy based on paired-sample Wilcoxon signed-rank test, corrected for multiple comparisons. Blue refers to digital delivery; orange refers to in-person delivery. ● Indicates outlier.

**Figure 2 ijerph-22-00220-f002:**
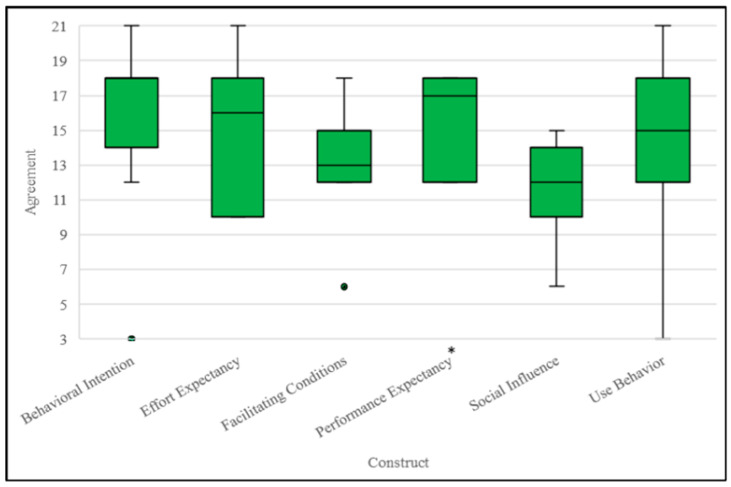
Box and whisker plots indicating client participants’ (*n* = 11) median UTAUT construct scores, first and third quartiles, and minimum and maximum scores. * = Significant difference (*p* < 0.05) between the median score and a reference score of 12 (total score of 3 questions asked based on Likert scale from 1 to 7) based on a one-sample Wilcoxon signed-rank test, corrected for multiple comparisons. ● Indicates an outlier.

**Table 1 ijerph-22-00220-t001:** Client participants’ (*n* = 11) median AQMH survey statistical analysis results.

AQMH Dimension	*p*-Value	z-Score	Effect Size
Ease of Use	0.201	−1.279	−0.386
Convenience	0.011 *	−2.536	−0.765
Acceptability	0.109	−1.604	−0.484
Practicality	0.011 *	−2.539	−0.767
Accessibility	0.012 *	−2.527	−0.762
Appropriateness	0.131	−1.511	−0.456
Effectiveness	0.18	−1.342	−0.405
Efficiency	0.014 *	−2.546	−0.768
Safety	0.44	−0.679	−0.205
Fit	0.109	−1.342	−0.405

* = Significant difference (*p* < 0.05) between median scores for digital delivery vs. in-person therapy based on paired-sample Wilcoxon signed-rank test, corrected for multiple comparisons. Effect size based on the following formula: r = Z/√N.

**Table 2 ijerph-22-00220-t002:** Client participants’ (*n* = 11) median UTAUT survey statistical analysis results.

UTAUT Dimension	*p*-Value	z-Score	Effect Size
Performance Expectancy	0.011 *	2.546	0.768
Effort Expectancy	0.032	2.140	0.645
Social Influence	0.67	−0.426	−0.128
Facilitating Conditions	0.368	0.900	0.271
Behavioral Intention	0.057	1.904	0.574
Use Behavior	0.182	1.335	0.403

* = Significant difference (*p* < 0.05) between the median score and a reference score of 12 (total score of 3 questions asked based on Likert scale from 1 to 7) based on a one-sample Wilcoxon signed-rank test, corrected for multiple comparisons. Effect size based on the following formula: r = Z/√N.

**Table 3 ijerph-22-00220-t003:** Summary of interview results.

Theme	Brief Description
Creating Connection While Online	Feeling ready to engage with digitally delivered psychotherapies utilized for trauma-affected populations aided in developing a strong therapeutic alliance with their clinician, improving participants’ therapeutic experience.
Improved Access to Care	Client participants agreed that the most crucial benefit of digital delivery was that it increased the accessibility of psychotherapies utilized for trauma-affected populations.
Differing Experiences Working Digitally	Some participants felt that receiving digitally delivered care made them less anxious than receiving in-person care, while others felt that receiving digitally delivered care made it more difficult to engage with therapy.
Difficulties with Working Digitally	Participants raised concerns regarding accessing resources required to attend digitally delivered sessions and worries about their security and privacy while using DMHI.
Continuing to Improve Access to Care	Participants provided several recommendations for integrating digital delivery into psychotherapy care for trauma-affected populations, including the expansion of hybrid care.

## Data Availability

The original contributions presented in this study are included in the article; further inquiries can be directed to the corresponding author.

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
