# Peer review of "Pilot Study Exploring the Perspectives of Canadian Clients Who Received Digitally Delivered Psychotherapies Utilized for Trauma-Affected Populations"

_ijerph, 2025, doi:10.3390/ijerph22020220_

Round 1

Reviewer 1 Report

Comments and Suggestions for Authors

Introduction Recommendations: 1-The introduction should explicitly state the unique traits of the Canadian population that previous studies, especially those in the United States, have overlooked. This includes cultural differences, specific healthcare needs, and unique challenges Canadians face in accessing or utilizing digital therapies. Emphasizing these aspects will strengthen the study's rationale. 2- Although evidence supports the effectiveness of digital therapies in the U.S., there is a lack of research on Canadian experiences with these interventions. This study should highlight how it fills this gap, underscoring its contribution to the field and the importance of examining this under-researched area. 

In the Method section: Please report the validity and reliability of the instruments.

Author Response

Thank you for your insightful comments. Our responses are in green font below.

Introduction Recommendations: 

1-The introduction should explicitly state the unique traits of the Canadian population that previous studies, especially those in the United States, have overlooked. This includes cultural differences, specific healthcare needs, and unique challenges Canadians face in accessing or utilizing digital therapies. Emphasizing these aspects will strengthen the study's rationale. 2-Although evidence supports the effectiveness of digital therapies in the U.S., there is a lack of research on Canadian experiences with these interventions. This study should highlight how it fills this gap, underscoring its contribution to the field and the importance of examining this under-researched area. 

Thank you for these suggestions. We have added the following to the manuscript to address these points:

“Multiple differences between Canadian and American client populations, including differing social characteristics (e.g., different cultural backgrounds), healthcare needs, and geographical challenges, may substantially impact how these populations access and utilize DMHI.” [page 3, lines 119-122] 

3-In the Method section: Please report the validity and reliability of the instruments.

All measures were co-created by members of the research team, including diverse opinions of experts from a variety of disciplines (computational psychiatry, psychology, psychiatry, nursing, rehabilitation medicine), which aided in increasing the validity and reliability of the instruments. However, as we created all instruments used in this study, we cannot report on their validity and reliability directly. 

Reviewer 2 Report

Comments and Suggestions for Authors

I found this pilot study to be really insightful in terms of understanding how digitally delivered psychotherapies can help trauma-affected populations, particularly in the context of Canadian military and public safety personnel. The mixed-methods approach was a great way to combine quantitative data with qualitative themes, which added depth to the findings. However, there are some areas where the manuscript could be improved to make it even more impactful and applicable.

Firstly, I think it would be helpful to mention the small sample size in the abstract, just to set expectations for readers. Additionally, including some relevant keywords like "digital mental health interventions" and "mixed-methods research" could make the manuscript more discoverable.

In terms of the introduction, I think it would be beneficial to provide a broader context by discussing global trends in digital mental health interventions and comparing Canadian-specific challenges with international findings. This would help generalize the study's relevance and make it more interesting to a wider audience. It might also be worth referencing some recent literature on emerging therapeutic approaches like Neurofeedback combined with Trauma-Informed Motivational Interviewing (doi: 10.20944/preprints202410.1594.v1).

Regarding the methods section, I appreciate that the authors acknowledged the small sample size as a limitation. However, I think it would be helpful to provide more detail on how this limitation was mitigated, such as through rigorous qualitative analysis. It's also worth explaining why certain participants were excluded, such as non-English-speaking individuals or those without access to technology.

In terms of the results, I think presenting confidence intervals or effect sizes alongside p-values would give a clearer picture of the findings' robustness. It might also be worth exploring potential reasons and implications for non-significant findings in the AQMH survey. The qualitative themes were well-structured, but including more participant quotes could provide richer context and bring the themes to life.

The discussion section could explore ethical concerns deeper, like maintaining privacy during virtual therapy in shared living spaces. It might also be worth highlighting the potential of hybrid models that combine in-person and digital therapies. Comparing the findings with relevant studies on topics like VR-based therapies (https://doi.org/10.1016/j.ssmmh.2024.100351) could situate the study within the broader field of digital mental health research.

In terms of limitations, I think it's great that the authors noted socioeconomic bias in their sample. However, proposing strategies to address this issue in future research could strengthen its practical implications. For example, providing devices or training for participants from diverse socioeconomic backgrounds could help mitigate technological barriers.

Finally, I think advocating for hybrid care models that address privacy and technological barriers would provide actionable insights for policymakers and clinicians is essential if we want this work translated into meaningful impacts.

Author Response

Thank you for reviewing our manuscript. Our responses to your comments are in green font below.

I found this pilot study to be really insightful in terms of understanding how digitally delivered psychotherapies can help trauma-affected populations, particularly in the context of Canadian military and public safety personnel. The mixed-methods approach was a great way to combine quantitative data with qualitative themes, which added depth to the findings. However, there are some areas where the manuscript could be improved to make it even more impactful and applicable.

Thank you for your kind remarks.

Firstly, I think it would be helpful to mention the small sample size in the abstract, just to set expectations for readers. Additionally, including some relevant keywords like "digital mental health interventions" and "mixed-methods research" could make the manuscript more discoverable.

Thank you for this suggestion. We have added the following to the abstract: 

“Participants indicated unique advantages of digital delivery, including the increased accessibility of treatment, cost effectiveness, and more efficient use of resources, although the small sample size limits the generalizability of our findings.” [page 1, lines 36-38] 

We have also added “digital mental health interventions” and “mixed-methods research” as keywords. 

In terms of the introduction, I think it would be beneficial to provide a broader context by discussing global trends in digital mental health interventions and comparing Canadian-specific challenges with international findings. This would help generalize the study's relevance and make it more interesting to a wider audience. It might also be worth referencing some recent literature on emerging therapeutic approaches like Neurofeedback combined with Trauma-Informed Motivational Interviewing (doi: 10.20944/preprints202410.1594.v1).

Thank you for this suggestion. While we agree that contextualizing our study in the broader context of all DMHI, including virtual reality, is important, this begins to fall outside the scope of our study. In this pilot study we investigated the perceptions of clients that received digitally delivered synchronous therapist-delivered psychotherapeutic interventions specifically. As this is a pilot study, follow-up studies could be conducted to compare the use of digitally delivered synchronous therapist-delivered psychotherapeutic interventions and digitally delivered interventions such as the Neurofeedback combined with Trauma-Informed Motivational Interviewing, as suggested by the reviewer. We have therefore decided to include the following into the manuscript:

“The use of digitally delivered therapies is becoming increasingly common and rapidly evolving to include virtual reality-based interventions [23,24]. Against this backdrop we conducted this pilot study to investigate the perceptions of synchronous therapist-delivered digitally delivered psychotherapies utilized for trauma affected populations, as reported by Canadian military members, Veterans, and PSP who have undergone these interventions.” [page 3, lines 123-128]

  1. Spytska, L. The use of virtual reality in the treatment of mental disorders such as phobias and post-traumatic stress disorder. SSM Mental Health. 2024, 6(3), 100351. doi: 10.1016/j.ssmmh.2024.100351
  2. Triscari, S.; Casu, M.; Uccelli, E.; Vitale, N. M.; Rapisarda, V.; Fakhrou, A.; Caponnetto, P. Virtual Reality Neurofeedback Training combined with Trauma-Informed Motivational Interviewing for the Treatment of Post-Traumatic Stress Disorder. Preprints 2024, 2024101594. https://doi.org/10.20944/preprints202410.1594.v1

We briefly speak about global vs Canadian trends in the introduction and discussion. We have decided to add the following to the manuscript to further speak to global trends in digital mental health research: 

“Such studies have been comparatively rare, however, with much of the extant DMHI literature primarily focusing on the use of DMHI in civilian populations, such as the use of ICBT in civilian populations [21,22]. This has limited our ability to understand how the shift to digital delivery affected PPTE-focused treatments specifically.” [page 3, 106-110]

  1. Andrews, G.; Basu, A.; Cuijpers, P.; Craske, M.G.; McEvoy, P.; English, C.L.; Newby, J.M. Computer Therapy for the Anxiety and Depression Disorders is Effective, Acceptable and Practical Health Care: An Updated Meta-Analysis. J. Anxiety Disord. 2018, 55, 70–78.
  2. Carlbring, P.; Andersson, G.; Cuijpers, P.; Riper, H.; Hedman-Lagerlof, E. Internet-Based vs. Face-to-Face Cognitive Behavior Therapy for Psychiatric and Somatic Disorders, An Updated Systematic Review and Meta-Analysis. Cogn. Behav. Ther. 2018, 47, 1–18. 

Regarding the methods section, I appreciate that the authors acknowledged the small sample size as a limitation. However, I think it would be helpful to provide more detail on how this limitation was mitigated, such as through rigorous qualitative analysis. It's also worth explaining why certain participants were excluded, such as non-English-speaking individuals or those without access to technology.

Thank you for these suggestions. We address these points in the methods and limitations sections of the manuscript. 

“Non-English speaking individuals were excluded due to the research team’s limited capacity to conduct interviews and analyze data in multiple languages.“ [page 4, lines 175-177]

“Steps were taken in this mixed-methods study to mitigate the effects of the small sample size, including the use of rigorous and rich qualitative data collection to enhance the credibility and depth of findings. Purposive sampling was done to ensure diverse perspectives, while integrating qualitative and quantitative results strengthened interpretability. Comparing findings with existing literature and providing thick descriptions improved transferability, while transparently acknowledging limitations reinforced methodological rigor. While these strategies aided in mitigating sample size constraints and enhanced the study’s contribution, further research with larger sample sizes are needed to replicate the findings reported here and to explore potential moderating factors, such as differences related to gender, sex, and sexual orientation with regards to the acceptance of digitally delivered care.” [page 13, lines 583-593]

“As research activities took place over online mediums, those without access to technology were excluded due to feasibility reasons.” [page 13, lines 297-598]

In terms of the results, I think presenting confidence intervals or effect sizes alongside p-values would give a clearer picture of the findings' robustness. It might also be worth exploring potential reasons and implications for non-significant findings in the AQMH survey. The qualitative themes were well-structured, but including more participant quotes could provide richer context and bring the themes to life.

Thank you for this suggestion. We have added the z-scores and effect sizes to Tables 1 and 2 for clarity [pages 7 and 8]. Given the study sample size, we believe it would not be appropriate to explore potential reasons and implications for non-significant findings for the AQMH survey results. Such reasons could be explored in a future study with a larger sample size. 

We also unfortunately do not have any further quotes to add to the manuscript, which is a limitation of the sample size. We hope to conduct further interviews in the future to add richness to the results presented in this manuscript. 

The discussion section could explore ethical concerns deeper, like maintaining privacy during virtual therapy in shared living spaces. It might also be worth highlighting the potential of hybrid models that combine in-person and digital therapies. Comparing the findings with relevant studies on topics like VR-based therapies (https://doi.org/10.1016/j.ssmmh.2024.100351) could situate the study within the broader field of digital mental health research.

We appreciate this feedback. Both VR therapy and our study highlight the potential of technology to improve access to care, particularly for individuals facing geographical or mobility barriers. While VR therapy provides controlled, individualized exposure to treat phobias and PTSD, our study examines the broader experiences of working digitally, including both benefits and challenges. VR-based CBT focuses on structured symptom relief, whereas our findings emphasize the importance of creating a connection online and addressing difficulties with digital interactions. Additionally, while VR therapy research demonstrates clear therapeutic efficacy, our study underscores the need for ongoing improvements in digital care delivery to enhance user experiences. Together, these insights highlight the balance between leveraging technology for treatment and ensuring meaningful, effective human connection in digital mental health interventions.

We do note that hybrid care was brought up by participants as an option to improve treatment in the future. We have added the following to the discussion section to address the potential ethical issues of DMHI brought up by participants:

“A third concern shared by participants regarding DMHI was the potential lack of privacy and security while attending digitally delivered sessions. For example, some participants found it difficult to find an area in their home to attend psychotherapy sessions that was sufficiently removed from their families. This made it difficult for them to openly speak about their traumatic experiences, with their worries of conveying information regarding their traumas overtaking their ability to focus on their psychotherapy session. These ethical concerns substantially negatively impacted their therapeutic experience, making them feel disconnected from their session and unsure whether DMHI could provide an adequate replacement for in-person care. Further research is needed to better understand how to adequately address these ethical concerns.” [page 12, lines 550-559]

In terms of limitations, I think it's great that the authors noted socioeconomic bias in their sample. However, proposing strategies to address this issue in future research could strengthen its practical implications. For example, providing devices or training for participants from diverse socioeconomic backgrounds could help mitigate technological barriers.

Thank you for this recommendation. We have added the following to the limitations section:

The provision of technology to those living in rural centers or the expansion of training programs for clients from diverse socioeconomic backgrounds may aid in mitigating some of the currently identified barriers to accessing DMHI. Further research is needed to identify equitable and sustainable strategies for providing digitally delivered psychotherapies utilized for trauma affected populations which acceptably meet the needs of Canada’s varied populations.” [page 13, lines 613-618]

Finally, I think advocating for hybrid care models that address privacy and technological barriers would provide actionable insights for policymakers and clinicians is essential if we want this work translated into meaningful impacts.

We agree with this sentiment and speak to the potential of hybrid care at the end of the discussion section. We have added the following to the manuscript: 

“Client participants supported the increased and expanded usage of hybrid services, where they could attend digital or in-person sessions depending on their needs. Hybrid care models could be used to address some of the issues surrounding DMHI, including privacy and technology barriers, assisting in clients receiving the most optimal services for their mental healthcare needs. Moving forward it will be imperative that mental health clinicians are prepared and able to identify which clients with what symptoms are most appropriate for digital, in-person, or hybrid services, which could aid in reaching positive treatment outcomes. Future research should therefore prioritize identifying which individuals would benefit most from such interventions.” [pages 12-13, lines 560-572]

Reviewer 3 Report

Comments and Suggestions for Authors

Dear Authors, thank you for the opportunity to review your manuscript. It is a very interesting study that contributes to the evidence of the viability and acceptance of online treatments.

Here are some observations:

Maximum five keywords

INTRODUCTION

It is necessary to explain which and how the interventions used are technically.

Greater depth should be generated in the justification of the research. Please add more references based on studies about online therapeutic processes in general and then look for any that exist in relation to the topic addressed by this study.

RESULTS

In the section where the types of treatment they received are described, to clarify how many of them completed the therapeutic process. It would also be interesting to establish the length of therapy they received.

In table 1, 2, the description of the importance should go below the table, not in the title of the table.

In the qualitative analysis, is it possible to make a table with the main driving ideas that appeared among the participants?

DISCUSSION

The discussion does not discuss how these results can be interpreted from the perspective of trauma or what the relevance of this research is in trauma specifically. It is necessary to delve deeper into the psychological and psychopathological aspects and the implications that this study has for future professionals who decide to work online on these topics.

Author Response

Thank you for reviewing our manuscript. Our response to your comments are in green font below.

Dear Authors, thank you for the opportunity to review your manuscript. It is a very interesting study that contributes to the evidence of the viability and acceptance of online treatments.

Thank you for your thoughtful feedback on this manuscript.

Here are some observations:

Maximum five keywords

Thank you for addressing this error. We have changed the keywords to the following: 

Digital mental health interventions, mixed-methods research, trauma-focused psychotherapy, military members and Veterans, and public-safety personnel 

INTRODUCTION

It is necessary to explain which and how the interventions used are technically.

Thank you for this feedback. Instead of focusing on the technical aspects of how specific interventions were digitally offered,  we sought to better understand the broader perspectives and experiences of clients with the goal of assessing the implementation of digital mental health interventions. This unfortunately means that we are unable to directly speak on the technical aspects of how specific psychotherapeutic interventions are provided digitally. We were, however, able to capture whether clients felt that digitally delivered interventions were usable and offered similar quality of care as in-person services through the survey measures we used.

Greater depth should be generated in the justification of the research. Please add more references based on studies about online therapeutic processes in general and then look for any that exist in relation to the topic addressed by this study.

Thank you for this suggestion. We have added the following to the introduction to better justify our research: 

“Such studies have been comparatively rare, however, with much of the extant DMHI literature primarily focusing on the use of DMHI in civilian populations, such as the use of ICBT in civilian populations [21,22]. This has limited our ability to understand how the shift to digital delivery affected PPTE-focused treatments specifically.” [page 3, 106-110]

  1. Andrews, G.; Basu, A.; Cuijpers, P.; Craske, M.G.; McEvoy, P.; English, C.L.; Newby, J.M. Computer Therapy for the Anxiety and Depression Disorders is Effective, Acceptable and Practical Health Care: An Updated Meta-Analysis. J. Anxiety Disord. 2018, 55, 70–78.
  2. Carlbring, P.; Andersson, G.; Cuijpers, P.; Riper, H.; Hedman-Lagerlof, E. Internet-Based vs. Face-to-Face Cognitive Behavior Therapy for Psychiatric and Somatic Disorders, An Updated Systematic Review and Meta-Analysis. Cogn. Behav. Ther. 2018, 47, 1–18. 

RESULTS

In the section where the types of treatment they received are described, to clarify how many of them completed the therapeutic process. It would also be interesting to establish the length of therapy they received.

Thank you for your comment. We unfortunately did not collect this data during this pilot study and recognize this is a limitation of our findings. We have added the following to address this in the limitations section: 

“A fourth limitation of this study was that we were able to only collect limited information regarding the specific psychotherapeutic interventions client participants had received. Future research should prioritize collecting data such as specific interventions received and the length of psychotherapy attendance to clarify whether participants were completing the therapeutic process or not.” [pages 13-14, lines 619-637]

In table 1, 2, the description of the importance should go below the table, not in the title of the table.

Thank you for this suggestion. We have moved these descriptions below the tables. [pages 7 and 8]

In the qualitative analysis, is it possible to make a table with the main driving ideas that appeared among the participants?

Thank you for this suggestion, we have added the following table to the manuscript. [Pages 8-9]

Table 3. Summary of interview results. 

Theme

Brief Description 

Creating Connection While Online

Feeling ready to engage with digitally delivered psychotherapies utilized for trauma affected populations aided in developing a strong therapeutic alliance with their clinician, improving participants’ therapeutic experience.

Improved Access to Care

Client participants agreed that the most crucial benefit of digital delivery was that it increased the accessibility of psychotherapies utilized for trauma affected populations. 

Differing Experiences Working Digitally

Some participants felt that receiving digitally delivered care made them less anxious than receiving in-person care, while others felt that receiving digitally delivered care made it more difficult to engage with therapy. 

Difficulties with Working Digitally

Participants raised concerns regarding accessing resources required to attend digitally delivered sessions and worries about their security and privacy while using DMHI.

Continuing to Improve Access to Care

Participants provided several recommendations for integrating digital delivery into psychotherapy care for trauma affected populations, including the expansion of hybrid care. 

DISCUSSION

The discussion does not discuss how these results can be interpreted from the perspective of trauma or what the relevance of this research is in trauma specifically. It is necessary to delve deeper into the psychological and psychopathological aspects and the implications that this study has for future professionals who decide to work online on these topics.

Thank you for this thoughtful comment. Given our investigation was focused on the implementation of DMHI rather than the psychological aspects of digital delivery, we feel we would be limited in speaking to the psychological and psychopathological aspects of trauma-focused care based on our current data. As this is a pilot study, future research in this area could focus more heavily on these topics, which we acknowledge are incredibly important parts of the therapeutic process and would be of interest to mental healthcare professionals working in this area. 

Reviewer 4 Report

Comments and Suggestions for Authors

Excellent and highly relevant article--especially in our post-covid reality. Well written and articulated. Addresses limitations adequately. 

One major limitation is not knowing how participants' previous experiences with trauma-based therapy (either in-person or digital) influenced or biased the results of this study. In line 420 there is mention that digitally delivered psychotherapies appeared to offer "similar treatment quality as in person delivery" but not all participants experienced both modalities. And some have but they didn't do a side by side comparison. Also how is "quality" measured. There is no mention of measured clinical outcomes for example (improved scores on a PCL-5 for example). 

Could variation in clinical practice between the various treatment centres that you recruited from, or variation in clinical outcomes between these clinics/clinicians influence patient perception of "quality" or skew their perceived preference. For example,  did participants prefer digital or in-person therapy becuase they achieved better symptom management? 

Another interesting correlate to explore in future research might be to assess clinician/provider perspectives in addition to client experiences and see if these align.

Regarding assessment of the therapeutic relationship (line 472) --future consideration might be to include scales that measure patient perceptions of therapeutic alliance such as the BR-WAI.  https://sphsoutcomes.net/sites/default/files/users/user824/BriefRevisedWorkingAllianceInv.pdf

In addition to the demographic variables mentioned in your limitation section lines 482-493 and line 508 (and areas for future research), one consideration that will be relevant to health system planners is to explore variation in trauma resource availability between rural and urban centers (e.g. distance to travel) and whether this influences clients' perceived preferences for accessing digital verses in-person care.

Author Response

Thank you for reviewing our manuscript. Our responses to your comments are in green font below.

Excellent and highly relevant article--especially in our post-covid reality. Well written and articulated. Addresses limitations adequately. 

Thank you for your thoughtful review of our manuscript. We appreciate your comments that will help us make this manuscript better. 

One major limitation is not knowing how participants' previous experiences with trauma-based therapy (either in-person or digital) influenced or biased the results of this study. In line 420 there is mention that digitally delivered psychotherapies appeared to offer "similar treatment quality as in person delivery" but not all participants experienced both modalities. And some have but they didn't do a side by side comparison. Also how is "quality" measured. There is no mention of measured clinical outcomes for example (improved scores on a PCL-5 for example). 

Yes, we agree that this is a limitation, however this speaks to the power of digital delivery - those who had not used psychotherapy previously were not accessing treatment. 

We measured treatment quality through the Alberta Quality Matrix of Health survey, as each domain of the matrix is related to healthcare quality. While we recognize that we did not use specific instruments to measure clinical outcomes, we believe that collecting the perspectives of clients provides important context.   

Could variation in clinical practice between the various treatment centres that you recruited from, or variation in clinical outcomes between these clinics/clinicians influence patient perception of "quality" or skew their perceived preference. For example, did participants prefer digital or in-person therapy because they achieved better symptom management? 

Thank you for this comment. This is a possibility, and a potential limitation of our recruitment strategy. This is a factor that we would like to explore further in follow up research to this pilot study. 

Another interesting correlate to explore in future research might be to assess clinician/provider perspectives in addition to client experiences and see if these align.

We agree that this is an interesting correlate to explore. We have captured clinician perspectives in a separate publication (https://doi.org/10.3390/ijerph22010081). The views of clinicians aligned with the views of clients in this pilot study, with clinicians also supporting the continued use of digital mental health interventions. 

Yap, S.; Allen, R.R.; Bright, K.S.; Brown, M.R.G.; Burback, L.; Hayward, J.; Winkler, O.; Wells, K.; Jones, C.; Sevigny, P.R.; et al. Exploring the Perspectives of Canadian Clinicians Regarding Digitally Delivered Psychotherapies Utilized for Trauma-Affected Populations. Int. J. Environ. Res. Public Health 2025, 22, 81. https://doi.org/10.3390/ijerph22010081

Regarding assessment of the therapeutic relationship (line 472) --future consideration might be to include scales that measure patient perceptions of therapeutic alliance such as the BR-WAI.  https://sphsoutcomes.net/sites/default/files/users/user824/BriefRevisedWorkingAllianceInv.pdf

Thank you for this suggestion. We believe that adding this would be very prudent. As such we have added the following to the manuscript:

Some client participants expressed difficulties in forming a strong therapeutic relationship and emotional connection with their clinician, in line with previous research that has indicated that individuals who have experienced interpersonal trauma may have difficulties developing a strong therapeutic alliance [41]. Future research could consider collecting longitudinal data on perceptions of the therapeutic alliance using instruments, such as the Brief Revised Working Alliance [42], to track how perceptions of the therapeutic alliance change over time.” [page 12, lines 542-549]   

  1. Lawson, D.M.; Skidmore, S.T.; Akay‐Sullivan, S. The influence of trauma symptoms on the therapeutic alliance across treatment. J. Couns. Dev. 2020, 98(1), 29–40. doi: 10.1002/jcad.12297.
  2. Mallinckrodt, B.; Tekie, Y.T. Revision of the Working Alliance Inventory and Development of a Brief Revised Version Guided by Item Response Theory. Department of Psychology, University of Tennessee, Knoxville, TN, USA. 2015. 

In addition to the demographic variables mentioned in your limitation section lines 482-493 and line 508 (and areas for future research), one consideration that will be relevant to health system planners is to explore variation in trauma resource availability between rural and urban centers (e.g. distance to travel) and whether this influences clients' perceived preferences for accessing digital verses in-person care.

Thank you for bringing this consideration to our attention. This is a relevant factor to the potential usage of digital versus in-person care. We have therefore added the following to the manuscript:

“Differences in geographic location may substantially influence the willingness of individuals using digital or in-person care. For example, the variability in the availability of TFP resources between rural and urban centers may affect clients’ perceived preferences for using digitally delivered or in-person services.” [page 13, lines 609-613]

Round 2

Reviewer 2 Report

Comments and Suggestions for Authors

The revised manuscript has made significant progress in addressing the major concerns of the previous review and demonstrates a high level of rigour and improvement. However, there are some minor suggestions that could further improve the clarity and consistency of the paper. Here are my recommendations:

In the Methods section, a bit more detail on how privacy and security concerns were managed during digital sessions would be helpful. For example, you could discuss the use of Zoom encryption and data anonymization to ensure the safety and confidentiality of participants.

Also, ensure that p-values are reported consistently (e.g., "p = 0.011" vs. "p=0.011") and use italics for *p*. 

Finally, to strengthen the translational relevance of the study, it would be beneficial to briefly discuss practical steps for implementing hybrid care models in the Discussion section. This could include clinician training and the infrastructure needed to support such models.

Author Response

Thank you for the additional feedback on our manuscript, it is greatly appreciated. Our responses to your comments are in green text below.

The revised manuscript has made significant progress in addressing the major concerns of the previous review and demonstrates a high level of rigour and improvement. However, there are some minor suggestions that could further improve the clarity and consistency of the paper. Here are my recommendations:

In the Methods section, a bit more detail on how privacy and security concerns were managed during digital sessions would be helpful. For example, you could discuss the use of Zoom encryption and data anonymization to ensure the safety and confidentiality of participants.

Thank you for this suggestion. We have added the following to the manuscript:

All Zoom interviews were conducted using Zoom Business set up with end-to-end encryption and geolocation limited to servers within Canada.” [page 5, line 257-259] 

“Participants were given the option to turn their cameras off and change their Zoom display name during the interview to maintain their anonymity.” [page 5, lines 268-270] 

“This involved removing names and contact information.” [page 5, lines 273-274]  

Video-recorded interviews were transcribed using the automated transcription function in Adobe Premiere Pro, running locally on a secure University of Alberta computer. Transcription accuracy was checked by a research team member (SY or RW), anonymized, and thematically analyzed both deductively and inductively following an iterative process [37]. Research team members were blinded to the data during thematic analysis” [page 6, lines 296-300]    

Also, ensure that p-values are reported consistently (e.g., "p = 0.011" vs. "p=0.011") and use italics for *p*. 

Thank you for catching this error. We have fixed the formatting of how p-values are reported throughout the manuscript (added italics, p-values reported as p=0.011.) [pages 6 and  8] 

Finally, to strengthen the translational relevance of the study, it would be beneficial to briefly discuss practical steps for implementing hybrid care models in the Discussion section. This could include clinician training and the infrastructure needed to support such models.

Thank you for your insightful comment. We have added the following to the manuscript:

Practical steps, such as the expansion of clinical training programs focused on providing hybrid care and installation of accessible and reliable infrastructure for therapists and clients, can be taken to ensure the successful implementation of hybrid care models.”  [page 13, lines 594-597]